# Cohort profile: DOLORisk Dundee: a longitudinal study of chronic neuropathic pain

Harry L. Hébert [1], Abirami Veluchamy,[1,2] Georgios Baskozos [3]
Francesca Fardo,[4] Dimitri M. L. Van Ryckeghem,[5,6,7] Mathilde M. V. Pascal [3]
Claire Jones,[8] Keith Milburn,[8] Ewan R. Pearson,[2] Geert Crombez,[5]
David L. H. Bennett,[3] Weihua Meng,[1] Colin N. A. Palmer,[2] Blair H. Smith[1]

For numbered affiliations see end of article.

**Correspondence to**
Professor Blair H. Smith;
b.h.smith@dundee.ac.uk

## ABSTRACT

**Purpose** Neuropathic pain is a common disorder of the somatosensory system that affects 7%–10% of the general population. The disorder places a large social and economic burden on patients as well as healthcare services. However, not everyone with a relevant underlying aetiology develops corresponding pain. DOLORisk Dundee, a European Union-funded cohort, part of the multicentre DOLORisk consortium, was set up to increase current understanding of this variation in onset. In particular, the cohort will allow exploration of psychosocial, clinical and genetic predictors of neuropathic pain onset.

**Participants** DOLORisk Dundee has been constructed by rephenotyping two pre-existing Scottish population cohorts for neuropathic pain using a standardised 'core' study protocol: Genetics of Diabetes Audit and Research in Tayside Scotland (GoDARTS) (n=5236) consisting of predominantly type 2 diabetics from the Tayside region, and Generation Scotland: Scottish Family Health Study (GS:SFHS; n=20 221). Rephenotyping was conducted in two phases: a baseline postal survey and a combined postal and online follow-up survey. DOLORisk Dundee consists of 9155 participants (GoDARTS=1915; GS:SFHS=7240) who responded to the baseline survey, of which 6338 (69.2%; GoDARTS=1046; GS:SFHS=5292) also responded to the follow-up survey (18 months later).

**Findings to date** At baseline, the proportion of those with chronic neuropathic pain (Douleur Neuropathique en 4 Questions questionnaire score ≥3, duration ≥3 months) was 30.5% in GoDARTS and 14.2% in Generation Scotland. Electronic record linkage enables large scale genetic association studies to be conducted and risk models have been constructed for neuropathic pain.

**Future plans** The cohort is being maintained by an access committee, through which collaborations are encouraged. Details of how to do this will be available on the study website (http://dolorisk.eu/). Further follow-up surveys of the cohort are planned and funding applications are being prepared to this effect. This will be conducted in harmony with similar pain rephenotyping of UK Biobank.

## INTRODUCTION

Chronic pain, usually defined as pain lasting more than 3 months, can be either nociceptive or neuropathic (though these may

### Strengths and limitations of this study

► DOLORisk Dundee is unique in enabling longitudinal population studies of neuropathic pain and related chronic pain traits.

► It is the largest longitudinal cohort of neuropathic pain to date providing increased power to conduct epidemiological and genetic analyses.

► Use of a 'core' study protocol allows easy collaboration and comparison with other cohorts within the consortium as well as external cohorts such as UK Biobank, which is rephenotyping for neuropathic pain using a similar protocol.

► Dundee has created a bespoke database in which all participant data created within the DOLORisk consortium is held and is available subject to access committee approval.

► DOLORisk Dundee has common limitations associated with epidemiological studies, including loss to follow-up, and there was incomplete response to questions on aetiology and location of worst pain.

overlap). Neuropathic Pain (NP) is defined as 'pain that arises as a direct consequence of a lesion or disease affecting the somatosensory system'[1] and affects approximately 20% of people with chronic pain.[2] NP affects 7%–10% of the general population[3] and is associated with a set of unpleasant characteristics including burning, pins and needles and electric shock-like sensations. The disorder is associated with poor quality of life, with 17% of people with NP rating their experience as being 'worse than death'.[2] It also places a large economic burden on patients and healthcare services with reduced productivity and employability at work and increased use of primary care with increasing pain severity.[4] Predictably, NP is associated with greater anxiety, depression[5] and sleep disturbance[6] when compared with non-NP. Unfortunately, many common analgesics used to treat nociceptive pain, including opioids, are generally

ineffective in NP.[7] Also, first-line medications used to treat NP, including gabapentin, pregabalin, duloxetine and amitriptyline, achieve satisfactory pain relief in less than half of people treated.[7]

Examples of disorders underlying NP are diabetes mellitus (causing painful diabetic neuropathy), herpes zoster (causing postherpetic neuralgia) and multiple sclerosis (which can be associated with trigeminal neuralgia). Contributing physical traumas to the nervous system include spinal cord injury, surgical procedures and amputation (resulting in phantom limb pain). However, not everyone with the same underlying disease or lesion develops NP. This is well demonstrated in diabetic polyneuropathy where only 20%–30% of people develop pain.[3] The variation in onset and severity of NP is likely to be due to a combination of genetic, psychosocial, demographic and clinical factors. A recent twins study in the UK has estimated that genetic factors account for 37% of the variance in NP onset.[8] However, the precise identity, magnitude and interaction of these factors have been poorly characterised, with previous analyses suffering from the usual limitations associated with epidemiology studies, including small sample size, cross-sectional design and heterogeneity in case definition.[9 10] Nevertheless, detecting risk factors has the potential to improve prognosis and use of healthcare resources.

DOLORisk was set up in 2015 to understand the risk factors and determinants for NP.[11] Funded by the European Commission's Horizon 2020 programme, DOLORisk is a multicentre study with collaborators from 9 European countries assembling specific clinical cohorts to explore a common set of factors. The largest of these, DOLORisk Dundee, is based on two pre-existing population-based cohorts, and combines new phenotypic data, collected 18 months apart, with genomic data. This study addresses some of the shortcomings identified in previous studies,[12] and focusses on the development of a risk model for NP onset and progression. The data will be available to other researchers to address related questions.

## COHORT DESCRIPTION
### Participant sources
The recruitment flow for DOLORisk Dundee is provided in figure 1. The study makes use of two pre-existing Scottish genetic epidemiology cohorts; the Genetics of Diabetes Audit and Research in Tayside Scotland (GoDARTS) study[13] and Generation Scotland: The Scottish Family Health Study (GS:SFHS).[14] GoDARTS consists of 18306 participants drawn from the Tayside region of Scotland between 1998 and 2015 and 10,149 of which were recruited on the basis of their diagnosis of diabetes mellitus (most of which are type 2). The rest of the cohort was recruited as diabetes-free controls. GS:SFHS is a family-based cohort consisting of 24084 participants recruited via general practices across Scotland from 2006 to 2011. Demographic and lifestyle data were obtained in both cohorts via questionnaires, whereas basic health data was collected from clinical examinations. Participants also provided blood, urine or saliva samples for biochemical and whole-genome analysis. At the point of recruitment, participants provided informed consent for their baseline data to be linked pseudonymously to longitudinal National Health Service (NHS) medical records (including laboratory, prescribing, morbidity, hospital admissions and mortality data), and for these to be used in medical research. Furthermore, participants provided consent to being contacted again for future studies. Cohort profiles of both GoDARTS and GS:SFHS have been published, where more details of the recruitment process and statistics for the individual studies can be found.[13 14]

### Baseline survey
The eligibility criteria for the study are given in table 1. GS:SFHS collected information on the presence, site and severity of chronic pain, using validated questionnaires,[11] but did not screen for the presence of NP. As neither cohort held specific data on NP prior to the DOLORisk Dundee study, diabetic participants of GoDARTS (n=5236) and all participants of GS:SFHS (n=20221),

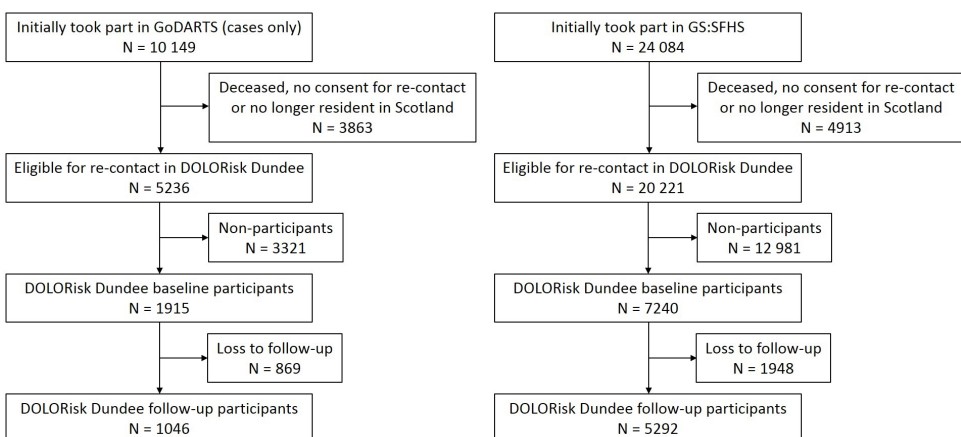

**Figure 1** Recruitment flow for DOLORisk Dundee. GoDARTS, Genetics of Diabetes Audit and Research in Tayside Scotland; GS-SFHS, Generation Scotland: Scottish Family Health Study.

**Table 1** Eligibility criteria for participation in DOLORisk Dundee

| Phase | Inclusion criteria | Exclusion criteria |
|---|---|---|
| Baseline | 1. Participants of Generation Scotland or GoDARTS | 1. Participants of GoDARTS who are 'controls' (diabetes free) |
| | 2. Provided consent to be recontacted about future studies as part of GoDARTS or GS:SFHS | |
| | 3. Living at the time of the DOLORisk study according to NHS medical records | |
| | 4. Resident in Scotland at the time of the DOLORisk study | |
| Follow-up | 1. Participated in the baseline survey | 1. Participants who had withdrawn from the study between baseline and follow-up |
| | 2. Provided additional consent to be recontacted about the follow-up survey | 2. Participants who had died between baseline and follow-up |
| | | 3. Participants who were no longer living in Scotland |

GoDARTS, Genetics of Diabetes Audit and Research in Tayside Scotland; GS:SFHS, Generation Scotland: Scottish Family Health Study; NHS, National Health Service.

who were consented, alive and resident in Scotland according to their NHS medical records, were contacted by post between May and December 2016 (figure 1). Each participant was invited to complete and return, by way of a prepaid envelope, a questionnaire on NP and other pain-related items (see below). Mailing packs also included the DOLORisk Dundee patient information leaflet (PIL) and invitation letter. Non-respondents were sent a reminder letter between November 2016 and April 2017, which also included a copy of the study PIL. The baseline survey was closed to respondents in September 2017. A total of 1915 people participated from GoDARTS by returning a completed questionnaire, a participation rate of 36.6% (1915/5236) and 7240 people returned a completed questionnaire from GS:SFHS, a participation

rate of 35.8% (7240/20 221). As part of the questionnaire, consent was obtained for further contact regarding a follow-up survey on NP and participants were invited to provide an email address for this purpose. The consent rate was 84.6% (1620/1915) in GoDARTS and 95.7% in GS:SFHS (6929/7240).

Sociodemographic statistics comparing participants and non-participants of DOLORisk Dundee from GoDARTS and GS:SFHS are presented in table 2. In GoDARTS participants were younger (71 years vs 73 years, p<0.01), had a higher proportion of males (61% vs 53%, p<0.01) and were from less socioeconomically deprived areas (living in most affluent category of the Scottish Index of Multiple Deprivation (SIMD) Quintile 1: 16% vs 23%, p<0.01) than non-participants, while ethnicity was similar

**Table 2** Baseline characteristic comparisons between DOLORisk Dundee participants and non-participants in GoDARTS and GS:SFHS

| | GoDARTS in DOLORisk dundee | | | GS:SFHS in DOLORisk dundee | | |
|---|---|---|---|---|---|---|
| | Participants | Non-participants | Overall | Participants | Non-participants | Overall |
| | (N=1915) | (N=3321) | (N=5236) | (N=7240) | (N=12 981) | (N=20 221) |
| Age (years)* | 71 | 73 | 72 | 60 | 51 | 55 |
| Gender (% male) | 61 | 53 | 56 | 39 | 41 | 40 |
| Ethnicity (% Caucasian) | 99.4 | 99.7 | 99.6 | 99.4 | 98.8 | 99 |
| SIMD (%) | | | | | | |
| 1 (most deprived) | 16 | 23 | 20 | 9 | 16 | 13 |
| 2 | 16 | 17 | 16 | 11 | 16 | 14 |
| 3 | 18 | 18 | 18 | 16 | 16 | 16 |
| 4 | 31 | 28 | 29 | 28 | 24 | 26 |
| 5 (least deprived) | 19 | 14 | 16 | 36 | 28 | 31 |

*Median values given for continuous variables.
GoDARTS, Genetics of Diabetes Audit and Research in Tayside and Scotland; GS:SFHS, Generation Scotland: Scottish Family Health Study; SIMD, Scottish Index of Multiple Deprivation.

in the two groups. Participants from GS:SFHS were older (60 years vs 51 years, p<0.01), had a lower proportion of males (39% vs 41%, p<0.01) and had a higher proportion of Caucasians (99.4% vs 98.8%, p<0.01), compared with non-participants, while participants were also less deprived than non-participants (SIMD quintile 1: 9% vs 16%, p<0.01).

### Follow-up Survey

Approximately 18 months after the DOLORisk Dundee baseline survey ended, participants from both GoDARTS (n=1460) and GS:SFHS (n=6657) who gave their consent to be contacted again, had not withdrawn in the meantime and were still alive, were invited to participate in a follow-up survey. The mortality rate of DOLORisk Dundee participants from GoDARTS in the time period between the baseline survey and the follow-up survey was 6.1% (116/1915) and in GS:SFHS was 0.5% (38/7240). Living participants who provided an email address in the baseline survey were sent an email containing a hyperlink to the follow-up survey website along with personal login credentials. The website also included a link to the study PIL. Simultaneously, living participants who provided consent for recontact but did not provide an email address were invited to participate in the follow-up survey by post. As in the baseline survey, mailings consisted of an invitation letter, questionnaire, study PIL and prepaid envelope. Participants of both the online and postal follow-up survey were sent reminders 3 weeks after the initial invitations. Invitations and reminders for the follow-up survey were sent between February and April 2019 and the survey was closed to respondents in June 2019. In GoDARTS, the response rate for the follow-up survey was 71.6% (1046/1460) and in GS:SFHS the response rate was 79.5% (5292/6657).

Table 3 describes the baseline sociodemographic characteristics in participants and non-participants (loss to follow-up, including those who were not sent a questionnaire due to non-consent or death) of the follow-up survey. In GoDARTS, participants were younger (69 years vs 72 years, p<0.01) than non-participants, while in GS:SFHS participants were older (61 years vs 58 years, p<0.01). Participants from GS:SFHS were also less deprived than non-participants (SIMD quintile 1: 8% vs 10%, p<0.01).

### Data collection

A summary of the data collected at baseline and follow-up and the completion rates are provided in table 4. All questionnaire mailing, returns and data entry/quality control were handled by the Health Informatics Centre at the University of Dundee, which is independent of the DOLORisk Dundee study team. This ensured that personal identifying information such as home and email addresses remained shielded from the research team. Questionnaire response data were made available to the study team via a secure bespoke database that was setup specifically for the DOLORisk study as a whole. Data were pseudonymised through the generation of study specific linkage IDs, which protects participant's identities and confidentiality while also ensuring that data can be linked across different datasets.

The questionnaire for the baseline survey follows the common DOLORisk 'core' protocol for phenotyping participants for NP, which was agreed on by all participating centres,[11] and based on pre-existing international consensus on NP phenotyping.[15] The items that were included in the questionnaire can be broadly split into two halves, with the first half consisting of psychosocial items and the second half consisting of pain phenotyping items.

**Table 3** DOLORisk Dundee loss to follow-up by baseline characteristics in GoDARTS and GS:SFHS

| | GoDARTS | | GS:SFHS | |
| --- | --- | --- | --- | --- |
| | Participants | Non-participants | Participants | Non-participants |
| | (N=1046) | (N=869)* | (N=5292) | (N=1948)* |
| Age (years)† | 69 | 72 | 61 | 58 |
| Gender (% male) | 62 | 59 | 39 | 39 |
| Ethnicity (% Caucasian) | 99.3 | 99.4 | 99.5 | 99.1 |
| SIMD (%) | | | | |
| 1 (most deprived) | 14 | 18 | 8 | 10 |
| 2 | 16 | 15 | 11 | 12 |
| 3 | 19 | 18 | 16 | 15 |
| 4 | 31 | 32 | 27 | 30 |
| 5 (least deprived) | 21 | 17 | 38 | 33 |

*Non-participant sample size includes those who were not sent a follow-up questionnaire (due to non-consent or death).
†Median values given for continuous variables.
GoDARTS, Genetics of Diabetes Audit and Research in Tayside and Scotland; GS:SFHS, Generation Scotland: Scottish Family Health Study; SIMD, Scottish Index of Multiple Deprivation.

**Table 4**  Summary of the data collected for the DOLORisk Dundee baseline and follow-up surveys

| Phase | Characteristic | Screening tool | Question number (GoDARTS/GS:SFHS) | Response rate (%)* GoDARTS (N=1915) | GS:SFHS (N=7240) |
|---|---|---|---|---|---|
| Baseline (Sept 2017) | Health-related quality of life | EQ-5D-5L | 1 | 90.3 | 95.1 |
| | Health-related quality of life | EQ-VAS | 2 | 95.7 | 97.4 |
| | Depression | PROMIS SF4a | 3a-d | 85.8 | 95.1 |
| | Anxiety | | 3e-f | 85.2 | 95.3 |
| | Sleep disturbance | | 4 | 79.8 | 90.6 |
| | Before the age of 18, have you ever experienced severe traumatic events? | NA | 5 | 91.9 | 97.1 |
| | Before the age of 18, have you ever stayed in hospital for a long period? | NA | 6 | 87.3 | 94.8 |
| | Extraversion | TIPI | 7a+f | 86.1 | 95.6 |
| | Agreeableness | | 7b+g | 87.4 | 96.1 |
| | Conscientiousness | | 7c+h | 86.9 | 96.0 |
| | Emotional stability | | 7d+i | 87.7 | 96.5 |
| | Open to new experiences | | 7e+j | 86.9 | 96.2 |
| | Mouth ulcers | NA | 8a | 91.3 | 96.8 |
| | Painful gums | NA | 8b | 89.3 | 96.2 |
| | Bleeding gums | NA | 8c | 89.0 | 96.5 |
| | Loose teeth | NA | 8d | 87.3 | 95.7 |
| | Toothache | NA | 8e | 88.4 | 95.9 |
| | Dentures | NA | 9 | 87.5 | 93.4 |
| | Ever regularly smoked tobacco? | NA | 10 | 95.7 | 99.7 |
| | If yes, age when started‡ | NA | 11 | 98.9 | 99.5 |
| | If yes, age when stopped or still smoking‡ | NA | 12 | 95.6 | 97.9 |
| | If yes, average no of cigars, cigarettes or grams of tobacco smoked in a week?‡ | NA | 13 | 92.3 | 96.6 |
| | How often do you currently drink alcohol? | NA | 14 | 95.6 | 99.6 |
| | On average how many pints of beer, 125 mL glasses of wine or 25 mL shots of spirit do you drink per week?§ | NA | 15 | 71.8 | 87.3 |
| | Pain Catastrophising | PCS | 16 | 86.7 | 96.2 |
| | Diabetic Peripheral Neuropathy | MNSI | 17–29/NA† | 82.8 | NA |
| | Currently troubled by pain or discomfort? | NA | 30/17 | 95.6 | 99.0 |
| | Currently taking pain medication? | NA | 31/18 | 95.5 | 98.7 |
| | Pain duration¶ | NA | 32/19 | 97.0 | 97.8 |
| | Location of any pain¶ | NA | 33a/20a | 94.4 | 96.2 |
| | Location of worst pain¶ | NA | 33b/20b | 60.0 | 73.4 |
| | Pain cause¶ | NA | 36/23 | 63.4 | 72.7 |
| | Neuropathic pain¶ | DN4 | 34-35/21-22 | 68.6 | 81.4 |
| | | S-LANSS | 38-44/25-31 | 77.2 | 92.5 |
| | 24 hour average pain severity¶ | BPI | 37/24 | 95.3 | 95.9 |
| | Pain severity¶ | CPG | 45-51/32-38 | 88.7 | 96.3 |
| | Consent for follow-up | NA | 52/39 | 84.6 | 95.7 |

Continued

**Table 4** Continued

| Phase | Characteristic | Screening tool | Question number (GoDARTS/GS:SFHS) | Response rate (%)* GoDARTS (N=1915) | Response rate (%)* GS:SFHS (N=7240) |
|---|---|---|---|---|---|
| Follow-up (April 2019) | Health-related quality of life | EQ-5D-5L | 1 | 96.9 | 98.9 |
| | Health-related quality of life | EQ-5D-VAS | 2 | 97.3 | 99.1 |
| | Depression | PROMIS SF4a | 3a-d | 90.2 | 96.2 |
| | Anxiety | | 3e-h | 89.3 | 96.0 |
| | Sleep disturbance | | 4 | 80.5 | 91.6 |
| | Pain Catastrophising | PCS | 5 | 89.4 | 95.6 |
| | Diabetic Peripheral Neuropathy | MNSI | 6–18/NA† | 86.7 | NA |
| | Currently troubled by pain or discomfort?** | NA | 19/6 | 99.3 | 99.8 |
| | Currently taking pain medication?** | NA | 20/7 | 99.2 | 99.8 |
| | Pain duration†† | NA | 21/8 | 97.9 | 98.1 |
| | Location of any pain†† | NA | 22a/9a | 94.7 | 98.6 |
| | Location of worst pain†† | NA | 22b/9b | 67.2 | 86.3 |
| | Neuropathic pain†† | DN4 | 23-24/10-11 | 70.4 | 82.1 |
| | | S-LANSS | 26-32/13-19 | 82.6 | 87.1 |
| | Pain cause†† | NA | 25/12 | 68.4 | 85.8 |
| | Pain severity†† | CPG | 33-39/20-26 | 91.3 | 96.4 |

Do you or your doctor think that this pain is caused by any of the following? (please tick one box only).
(1) A surgical operation more than 3 months ago (2) Back problems such as a slipped disc, back surgery or sciatica (3) Bowel or other abdominal or pelvic problems (4) Diabetes (5) Arthritis, rheumatism, or another joint problem (6) Cancer orcancer treatment, such as chemotherapy (7) Any type of neuralgia, neuropathy or nerve damage (including spinal cord injury) (8) Shingles (9) Multiple sclerosis (10) Muscle problems, such as spasms, strains, tension or tendonitis (11) Leg ulcers (12) Loss of a limb (13) Stroke (14) Other cause/unknown (please specify)
In the past 3 months; a) which of these pains have you had, b) which one of these pains bothered you the most?.
(1) Back pain (2) Neck or shoulder pain (3) Facial or dental pain (4) Headache (5) Stomach ache or abdominal pain (6) Pain in your arms (7) Pain in your hands (8) Chest pain (9) Pain in your hips (10) Pain in your legs or knees (11) Pain in your feet (12) Pain throughout your body (widespread pain) (13) Other pain (please specify).
*A characteristic is only considered complete if all questions that make up the characteristic are non-missing.
†Item only present in GoDARTS survey.
‡Item only answered if response to question 10 is "yes" (GoDARTS=1029/GS:SFHS=2792).
§Item only answered if response to question 14 is not "never" (GoDARTS=1300/GS:SFHS=6432).
¶Item only answered if response to either question 30/17 or 31/18 is 'yes' (GoDARTS=1276/GS:SFHS=4524).
**These items required a response before the online questionnaire could be submitted.
††Item only answered if response to either question 19/6 or 20/7 is "yes" (GoDARTS=702/GS:SFHS=3319)
.BPI, Brief Pain Inventory; CPG, Chronic Pain Grade; DN4, *Douleur Neuropathique en 4 Questions*; EQ-5D-5L, EuroQoL-five dimensions-five levels; EQ-VAS, EuroQoL-Visual Analogue Scale; GoDARTS, Genetics of Diabetes Audit and Research in Tayside and Scotland; GS:SFHS, Generation Scotland: Scottish Family Health Study; MNSI, Michigan Neuropathy Screening Instrument; NA, not applicable; PCS, Pain Catastrophising Scale; PROMIS, Patient-Reported Outcomes Measurement Information System; SF4a, short form four answers; S-LANSS, self-completed Leeds Assessment of Neuropathic Symptoms and Signs; TIPI, Ten Item Personality Inventory.;

For the psychosocial items, which were completed by all participants, validated questionnaires were used to assess health-related quality of life (EuroQoL-five dimensions five levels and Visual Analogue Scale),[16] depression,[17] anxiety,[17] sleep disturbance (all PROMIS SF-4a),[18] personality dimensions (Ten Item Personality Inventory)[19] and pain catastrophising (pain catastrophising scale; PCS).[20] A summary of the results from respondents of these psychosocial items are given in table 5, dichotomised by cohort. When compared with GS:SFHS, GoDARTS had a higher mean score in all of the psychological aspects (where a higher score denotes more of the concept being measured), including depression, anxiety, sleep disturbance and pain catastrophising, as well as having a lower mean score in the five personality traits (extraversion, agreeableness, conscientiousness, emotional stability and open to new experiences) and health-related quality of life.

Other factors such as adverse childhood experiences (ACEs), dental complications, and smoking and alcohol history were assessed by self-report items that were specifically designed for DOLORisk.

For the phenotyping items, participants initially answered two screening questions, on the presence of current pain ('Are you currently troubled by pain or discomfort, either all the time or on and off?') and medication intake ('Are you currently taking medications specifically to treat pain or discomfort?'). The first

**Table 5** Results from respondents in GoDARTS and GS:SFHS completing the pain and pain interference items

| Study | | N* | Mean | SD | Median | IQR | Theoretical range |
|---|---|---|---|---|---|---|---|
| GoDARTS | EQ-5D-5L | 1710 | 0.67 | 0.276 | 0.735 | 0.282 | –1.594 |
| | PROMIS SF-4a | | | | | | |
| | Depression | 1644 | 49.7 | 9.3 | 49 | 16.3 | 41.0–79.4 |
| | Anxiety | 1632 | 48.7 | 9.4 | 48 | 15.5 | 40.3–81.6 |
| | Sleep Disturbance | 1528 | 50.7 | 9.2 | 50.5 | 12.3 | 32.0–73.3 |
| | TIPI | | | | | | |
| | Extraversion | 1649 | 4 | 1.4 | 4 | 2 | 1.0–7.0 |
| | Agreeableness | 1673 | 5.1 | 1.2 | 5 | 2 | 1.0–7.0 |
| | Conscientiousness | 1665 | 5.4 | 1.3 | 5.5 | 2 | 1.0–7.0 |
| | Emotional stability | 1679 | 4.9 | 1.5 | 5 | 2.5 | 1.0–7.0 |
| | Open to new experiences | 1664 | 4.6 | 1.3 | 4.5 | 1.5 | 1.0–7.0 |
| | PCS | 1661 | 10.2 | 11.8 | 6 | 14 | 0–52 |
| | MNSI | 1586 | 2.6 | 2.5 | 2 | 4 | 0–13 |
| | DN4 | 875 | 2.2 | 2.1 | 2 | 4 | 0–7 |
| | S-LANSS | 985 | 7.2 | 7.2 | 5 | 13 | 0–24 |
| GS:SFHS | EQ-5D-5L | 6888 | 0.823 | 0.184 | 0.837 | 0.25 | –1.594 |
| | PROMIS SF-4a | | | | | | |
| | Depression | 6888 | 47.4 | 7.9 | 41 | 12.9 | 41.0–79.4 |
| | Anxiety | 6899 | 47.9 | 8.5 | 48 | 13.4 | 40.3–81.6 |
| | Sleep Disturbance | 6563 | 48.4 | 8.4 | 48.4 | 10.5 | 32.0–73.3 |
| | TIPI | | | | | | |
| | Extraversion | 6922 | 4.2 | 1.5 | 4 | 2.5 | 1.0–7.0 |
| | Agreeableness | 6959 | 5.4 | 1.1 | 5.5 | 2 | 1.0–7.0 |
| | Conscientiousness | 6950 | 5.8 | 1.1 | 6 | 2 | 1.0–7.0 |
| | Emotional stability | 6983 | 5.1 | 1.5 | 5.5 | 2.5 | 1.0–7.0 |
| | Open to new experiences | 6963 | 4.9 | 1.2 | 5 | 2 | 1.0–7.0 |
| | PCS | 6966 | 6.8 | 8.3 | 4 | 9 | 0–52 |
| | DN4 | 3681 | 1.2 | 1.6 | 1 | 2 | 0–7 |
| | S-LANSS | 4138 | 4.9 | 5.9 | 3 | 8 | 0–24 |

*Sample size only includes participants with non-missing responses for all questions that make up the item.
DN4, Douleur Neuropathique en 4 Questions; EQ-5D-5L, EuroQoL-five dimensions-five levels; GoDARTS, Genetics of Diabetes Audit and Research in Tayside Scotland; GS-SFHS, Generation Scotland: Scottish Family Health Study; IQR, interquartile range; MNSI, Michigan Neuropathy Screening Instrument; PCS, Pain Catastrophising Scale; PROMIS SF4a, Patient-Reported Outcomes Measurement Information System short form four answers; SD, standard deviation; S-LANSS, self-completed Leeds Assessment of Neuropathic Symptoms and Signs; TIPI, 10-Item Personality Inventory.

of these questions is validated to identify cases and is the same as used in the original GS:SFHS cohort recruitment.[21] The second question was added to allow inclusion of anyone whose pain was temporarily relieved by analgesics at the time of completing the questionnaire and may therefore answer 'no' to the preceding question. This has been used previously in a collaborative GWAS between GS:SFHS and 23andMe.[22] A positive response to either, or both of these questions meant participants were invited to complete further items on pain characterisation. Those who responded negatively to both questions were instructed to miss out the pain characterisation questions and to immediately complete the follow-up survey

consent section. Pain phenotyping items included the seven self-report items from the 'Douleur Neuropathique en 4 Questions' (DN4)[23] and the seven self-report items of the 'Leeds Assessment of Neuropathic Symptoms and Signs' (S-LANSS).[24] Both questionnaires are screening tools for NP and have been validated for use in populations with NP. Both screening tools have also been validated as self-report items and are therefore suitable for use in a postal/online survey setting. The DN4 showed 78% sensitivity and 81% specificity, while the S-LANSS had 74% sensitivity and 76% specificity when compared with clinical examination.[23 24] When compared with GS:SFHS, GoDARTS had a higher mean score in both the

**Table 6** Examples of data available through pre-existing collections or NHS record linkage as part of GoDARTS or GS:SFHS

| Group | Examples |
|---|---|
| Demographics | Age, gender, ethnicity, SIMD |
| Anthropometrics | Height, weight, waist |
| Clinical | Blood pressure, resting pulse |
| Lifestyle | Smoking, alcohol consumption, physical activity |
| Scottish Morbidity Records | Primary and secondary diagnoses |
| General Registrar's Office | Mortality data |
| Biochemistry | Glucose, cholesterol, HDL |
| Prescribing | Analgesics |
| Genetics | Genome-wide genotyping |

GoDARTS, Genetics of Diabetes Audit and Research in Tayside Scotland; GS:SFHS, Generation Scotland: Scottish Family Health Study; HDL, high-density lipoprotein; NHS, National Health Service; SIMD, Scottish Index of Multiple Deprivation.

DN4 and S-LANSS, potentially reflecting the older age of GoDARTS and the diabetic nature of the cohort (table 5). The Chronic Pain Grade was used to assess pain severity and disability.[25 26] In addition to these screening tools, the Michigan Neuropathy Screening Instrument (MNSI) (minus two questions that do not contribute to the scoring algorithm) was used to assess peripheral neuropathy in the diabetes specific GoDARTS cohort only.[27] Additional customised items assessed pain duration, pain location (by way of a checklist) and self-reported cause of pain (from a list of common pain aetiologies).

The follow-up survey used a truncated, but otherwise identical version of the baseline survey, with the items concerning ACEs, dental complications, personality dimensions, smoking and alcohol history being excluded.

In addition to the data collected through the two surveys, DOLORisk Dundee has access to routinely collected longitudinal NHS health data, available through both GoDARTS and GS:SFHS by way of electronic record linkage (table 6). This is made possible by the Community Health Index number, which is a unique number assigned to each individual on first registration with NHS Scotland. Examples of data available through electronic record linkage are comorbidity, mortality and community dispensed prescribing. The study also has access to pre-existing data collected as part of GoDARTS or GS:SFHS, including genome-wide genotyping data.

## Patient and public involvement

No patients or members of the public were directly involved in the design, conduct or reporting of this study. However, patient and public involvement was a major component in the establishment of GS:SFHS, including protocol development and principles of participation and

data access.[28] The results of the DOLORisk Dundee study will be disseminated to the wider patient community.

## Findings to date

As the follow-up stage of this study has only recently been completed, no papers have yet been published using the DOLORisk Dundee data. However, table 1 gives the descriptive characteristics for participants in the baseline survey and allows comparison of participants from GoDARTS and GS:SFHS. Participants from GoDARTS are older compared with participants from GS:SFHS (71 years vs 60 years), have a higher proportion of males (61% vs 39%) and are more socially deprived (SIMD 1; 16% vs 9%). Because GoDARTS and GS:SFHS differ in terms of participant recruitment, diabetes status and demographic characteristics, we present the two cohorts separately. In further research we intend to use them as discovery or validation sets.

The DOLORisk Dundee cohort will enable research into the identification of factors that predict NP onset, presence, progression, severity and remission, as well as allowing estimation of the prevalence (at baseline) and 18-month incidence. Using the pain data derived from the baseline questionnaires, we have found that the proportion with 'possible' chronic NP was 30.5% in GoDARTS and 14.2% in GS:SFHS (using the definition: pain duration ≥3 months and DN4 score ≥3). Table 7 shows the demographic descriptive statistics of participants from GoDARTS and GS:SFHS, comparing those with 'possible' chronic NP to those with chronic nociceptive pain (pain duration ≥3 months and DN4 score <3) and those with no pain. In both cohorts, participants with 'possible' chronic NP were younger, were less likely to be male and were more socially deprived than those in the other two groups.

In addition to the descriptive characteristics of DOLORisk Dundee, a genome-wide association study meta-analysis has been conducted using the aforementioned NP phenotyping data (combined with the UK Biobank cohort using a prescription–based phenotype). Similar genetic studies have been conducted for NP using a prescription-based phenotype in GoDARTS.[29 30] Separately, environmental risk models have been developed, incorporating demographic, clinical and biochemical predictors available through pre-existing records (table 6) and psychological and lifestyle data collected through the baseline survey. Both genetic and environmental predictors have been identified in previous studies of NP.[31] This will use similar methodologies employed in low back and postsurgical pain.[32 33] The results of these analyses will be submitted for publication in the near future. Eventually the environmental and genetic analyses will be combined through the use of polygenic and environment risk scores to produce a joint genetic and environmental model that will enable better understanding of the relative contributions of genetic and environmental factors to the variation in NP onset and progression. Polygenic risk scores have already been explored to good effect in type 2 diabetes.[34]

**Table 7** DOLORisk Dundee baseline characteristics according to pain phenotype in GoDARTS and GS:SFHS

| | GoDARTS | | | Gs:SFHS | | |
|---|---|---|---|---|---|---|
| | Neuropathic pain | Non-neuropathic pain | No pain | Neuropathic pain | Non-neuropathic pain | No pain |
| | (N=482)* | (N=461)* | (N=560)† | (N=932)* | (N=2484)* | (N=2642)† |
| Age (years)‡ | 69.5 | 71§ | 71§ | 59 | 61.0§ | 58.0§ |
| Gender (% male) | 54 | 58 | 69§ | 33 | 38§ | 42§ |
| Ethnicity (% Caucasian) | 99.4 | 99.3 | 99.3 | 99.6 | 99.5 | 99.2 |
| SIMD (%) | | | | | | |
| 1 (most deprived) | 22 | 14§ | 11§ | 16 | 7§ | 6§ |
| 2 | 15 | 15 | 14 | 16 | 11 | 9 |
| 3 | 17 | 18 | 16 | 15 | 17 | 16 |
| 4 | 29 | 30 | 35 | 26 | 29 | 28 |
| 5 (least deprived) | 17 | 22 | 24 | 26 | 37 | 40 |

Neuropathic pain and Non-neuropathic pain determined by scores greater and less than 3/7 on DN4, respectively.

No pain defined as a negative response to both chronic pain identification questions.

*Neuropathic pain and non-neuropathic pain determined by scores greater and less than 3/7 on DN4 respectively.

†No pain defined as a negative response to both chronic pain identification questions.

‡Median values given for continuous variables.

§P<0.05 (compared with neuropathic pain)

DN4, Douleur Neuropathique en 4 Questions; GoDARTS, Genetics of Diabetes Audit and Research in Tayside Scotland; GS:SFHS, Generation Scotland: Scottish Family Health Study; SIMD, Scottish Index of Multiple Deprivation.

## Strength and limitations

The main strength of DOLORisk Dundee lies in the size of the cohort, both in terms of the number of participants who responded to the questionnaire surveys (and provided pain data) as well as the amount of phenotypic data (both genetic and non-genetic) that are available for analyses. This makes DOLORisk Dundee one of the largest studies of NP to date, with statistical power to conduct detailed epidemiological and genetic epidemiological research and the capability of being augmented by other cohorts within the DOLORisk consortium. The ability to easily combine different cohorts within DOLORisk is derived from the use of the 'core' protocol, which is being used by all participating centres, and this will extend to other cohorts that are based on the same phenotyping consensus.[15] Furthermore, the availability of follow-up data on chronic pain at three time points and on NP at two time points allows longitudinal analysis to be conducted. The results of this study will enable validation in much larger cohorts and will help develop further areas of research in NP. One such cohort that this could apply to is UK Biobank, which has been rephenotyped for chronic pain, including NP, using a completely consistent phenotyping approach. Responses have been received from >165 000 participants in UK Biobank; these have not yet been analysed but will potentially include ~12 000 with NP, assuming a prevalence of 7%.[3] This would make UK Biobank ideal for replication studies and would easily overcome issues of the 'Winner's Curse', where the effect sizes of initial studies tend to be overestimated.[9]

This study has some limitations. While the response rate for most of the individual items was good, the response rate to some, including those for 'cause of pain' and 'location of worst pain', were lower (table 4). This means that the study has reduced power to investigate specific locations and aetiologies of NP, including painful diabetic neuropathy, postherpetic neuralgia and trigeminal neuralgia. While we attempted to obtain a broad array of common NP aetiologies, it was not possible to capture detailed clinical information on neurological and sensory phenotypes, through the medium of a self-complete questionnaire. However, the use of GoDARTS as a diabetic cohort will strengthen analysis relating to diabetic neuropathy, particularly with the use of the MNSI as a screening tool.

In retrospect, the presentation of the pain location questions (having a two column checklist of body locations side-by-side, with the first column for 'any pain' and the second column for 'worst pain') may have confused participants into ignoring the second question. A possible future solution could be to have them appearing consecutively rather than concurrently. Alternatively, the low response rate to pain that 'bothers you the most' could indicate that the question is difficult to answer.

Second, while the participants were asked about whether they were currently taking medications to treat pain, they were not asked about other forms of pain

treatment, such as neuromodulation therapy. These are important aspects when assessing pain outcomes and will be considered in future surveys.

Third, the survey uses self-report, which could be subject to recall or reporting bias. Fourth, participants from DOLORisk Dundee are likely those who are particularly engaged in this study, or medical research in general. In addition to this, participants of the baseline survey in GoDARTS were younger, were more likely to be male and to be less socially deprived than non-participants. Participants from GS:SFHS were older, were more likely to be females and Caucasians, in addition to being less socially deprived than non-participants. This suggests that DOLORisk Dundee may under represent certain sociodemographic categories and so should not be used to estimate disease prevalence.

## COLLABORATION

Collaborations with researchers interested in chronic (neuropathic) pain are encouraged. More information can be found on the DOLORisk website (http://dolorisk.eu/). An access committee will be set up beyond the current funding period to review data access and collaboration requests. Details will be made available on the study website once the access procedures have been decided.

**Author affiliations**
[1]Chronic Pain Research Group, Division of Population Health and Genomics, Mackenzie Building, Ninewells Hospital and Medical School, University of Dundee, Dundee, UK
[2]Pat Macpherson Centre for Pharmacogenetics and Pharmacogenomics, Division of Population Health and Genomics, Ninewells Hospital and Medical School, University of Dundee, Dundee, UK
[3]Neural Injury Group, Nuffield Department of Clinical Neuroscience, John Radcliffe Hospital, University of Oxford, Oxford, UK
[4]Danish Pain Research Center, Department of Clinical Medicine, Aarhus University, Aarhus, Denmark
[5]Department of Experimental-Clinical and Health Psychology, Faculty of Psychology and Educational Sciences, Ghent University, Gent, Belgium
[6]Section Experimental Health Psychology, Clinical Psychological Science, Departments, Faculty of Psychology and Neuroscience, Maastricht University, Maastricht, Netherlands
[7]Institute of Health and Behaviour, INSIDE, University of Luxembourg, Luxembourg, Luxembourg
[8]Health Informatics Centre, Ninewells Hospital and Medical School, University of Dundee, Dundee, UK

**Acknowledgements** We are grateful to all the families who took part from both GoDARTS and Generation Scotland, the general practitioners and the Scottish School of Primary Care for their help in recruiting them, and the whole GoDARTS and Generation Scotland teams, which includes interviewers, computer and laboratory technicians, clerical workers, research scientists, volunteers, managers, receptionists, healthcare assistants and nurses. We acknowledge the support of the Health Informatics Centre, University of Dundee for managing and supplying the anonymised data and NHS Tayside, the original data owner.

**Contributors** The concept and design of DOLORisk Dundee was developed by BHS, CNAP and WM, DLHB and GC, who obtained funding and all necessary approvals. Acquisition, curation and analysis of data relating to DOLORisk were conducted by HLH, AV, KM, CJ and MMVP. Collection and processing of pre-existing data were conducted by CNAP and ERP for GoDARTS and BHS for GS:SFHS. GB, FF, DMLVR and ERP provided methodological input on data analysis. HLH drafted the manuscript. All authors critically revised the manuscript for important intellectual content and approved the final version for submission.

**Funding** This work was supported by the European Union's Horizon 2020 research and innovation programme under grant agreement No 633491 (DOLORisk). BHS, CNAP, DLHB, GC and WM are members of the DOLORisk consortium and are partly supported by this grant. HLH and AV are supported by DOLORisk. Generation Scotland received core support from the Chief Scientist Office of the Scottish Government Health Directorates (CZD/16/6) and the Scottish Funding Council (HR03006). Genotyping of the GS:SFHS samples was carried out by the Genetics Core Laboratory at the Wellcome Trust Clinical Research Facility, Edinburgh, Scotland and was funded by the Medical Research Council UK and the Wellcome Trust (Wellcome Trust Strategic Award 'STratifying Resilience and Depression Longitudinally' (STRADL) Reference 104036/Z/14/Z).The Wellcome Trust United Kingdom Type 2 Diabetes Case-Control Collection (supporting GoDARTS) was funded by The Wellcome Trust (072960/Z/03/Z, 084726/Z/08/Z, 084727/Z/08/Z, 085475/Z/08/Z, and 085475/B/08/Z) and as part of the EU IMI-SUMMIT programme.

**Competing interests** None declared.

**Patient consent for publication** Not required.

**Ethics approval** Ethical approval for the DOLORisk Dundee study was obtained from the Yorkshire and The Humber-South Yorkshire Research Ethics Committee (reference: 15/YH/0285). Ethical approval was already in place for the pre-existing GoDARTS (053/04) and GS:SFHS (05/S1401/89) cohorts.

**Provenance and peer review** Not commissioned; externally peer reviewed.

**Data availability statement** Data are available on reasonable request. Researchers may request access to DOLORisk Dundee data by applying to the DOLORisk access committee (http://dolorisk.eu/). Further details on how to do this will be posted on the study website in due course.

**ORCID iDs**
Harry L. Hébert http://orcid.org/0000-0003-1753-6592
Georgios Baskozos http://orcid.org/0000-0001-9237-5878
Mathilde M. V. Pascal http://orcid.org/0000-0002-2841-6943

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
