## [Reviewer comments · BMJ Open]

ARTICLE DETAILS

TITLE (PROVISIONAL)	Cohort Profile: DOLORisk Dundee – A longitudinal study of chronic neuropathic pain
AUTHORS	Hebert, Harry; Veluchamy, Abirami; Baskozos, Georgios; Fardo, Francesca; Van Ryckeghem, Dimitri; Pascal, Mathilde; Jones, Claire; Milburn, Keith; Pearson, Ewan; Crombez, Geert; Bennett, David; Meng, W.; Palmer, Colin; Smith, Blair

VERSION 1 – REVIEW

REVIEWER	Michael Staudt Michigan Head & Spine Institute, USA
REVIEW RETURNED	13-Aug-2020

GENERAL COMMENTS	The current article outlines the creation of a neuropathic pain database, DOLORISK Dundee, using two pre-existing Scottish population cohorts (GoDARTS and GS:SFHS). The GoDARTS cohort focused primarily on diabetic patients, whereas patients in GS:SFHS were recruited from general practice. From these cohorts, all diabetic patients from GoDARTS and all patients from GS:SFHS were mailed a questionnaire on neuropathic pain and other pain-related items. Neuropathic pain specifically was assigned based on responses to the DN4 and S-LANSS questionnaires. As this is a descriptive and not investigative article, the results outline the population demographics of those participants who responded and provided consent, as well as characteristics of those patients who completed 18-month follow-up.  Neuropathic pain is common and debilitating, however our understanding of its pathophysiology and natural history remain limited. One of the primary limitations is the lack of large-scale demographic data that collates validated outcome metrics and genomic data. DOLORISK Dundee is an exciting venture as it has the potential to longitudinally study a large neuropathic pain patient population with linkage to genetic association studies. This will hopefully allow collaboration with and comparison to similar population-based databases.  As this is a descriptive report of the baseline database information without formal analysis, my comments are limited to the nature of data collection and the potential future applications.  1. My primary criticism is the lack of data regarding pain characterization and etiology. Neuropathic pain is diverse, and includes multiple different mononeuropathies and polyneuropathies (both symmetric and asymmetric). Despite a consistent approach to overall pain phenotyping, there appears to
--

be limited ability to specifically identify different neuropathic pain phenotypes. Diabetic neuropathy is not the same as CRPS which is not the same as lumbar radiculopathy, which can dilute interpretations following formal analysis. As neuropathic pain syndromes can be quite heterogeneous, this tends to be a problem in smaller studies – the DOLORISK Dundee database may have challenges of a different nature if interpretations are made based on overall “neuropathic pain”, which will limit generalizability to specific conditions. “Pain cause” is included in table 3, reported in approximately 2/3 of respondents, and it is mentioned that this is self-reported information from a list of common pain etiologies. It is important to include both this list of pain etiologies, as well as the specific breakdown in table 3.

2. In addition to challenges in assessment based on individual phenotype, there are also limitations regarding the broader classification of central vs peripheral neuropathic pain. Although data are limited, previous studies have shown that patients with central pain overall do worse with medical management compared to peripheral pain. By combining these groups in one neuropathic pain cohort, this potentially dilutes generalizability of future findings.

3. Conversely, having a large cohort of diabetic patients in GoDARTS can be seen as a strength when assessing diabetic neuropathy specifically, as up to a third of diabetic patients develop painful diabetic neuropathy. Again, although the nature of data collection does not mean that all patients in this cohort have painful diabetic neuropathy specifically, this would be an advantage in terms of analyzing demographic and genetic information in relation to treatment responses, and following long-term natural history.

4. The question regarding medication intake is quite rudimentary (“Are you currently taking medications specifically to treat pain or discomfort?”), and the response rate of 95% or greater is not surprising. However, this response could be interpreted as patients taking the occasional ibuprofen all the way to regular and supervised use of methadone. In patients with neuropathic pain it is important to understand the classes of pain medications used (i.e. opioids, antidepressants, anticonvulsants, etc), and for opioids the morphine equivalent dose.

5. I did not find any information regarding the use of surgery (i.e. neuromodulation) in the treatment of neuropathic pain. This is also important information to discern in the assessment of long-term treatment outcomes. Future surveys should incorporate relevant questions.

6. The authors should indicate how biochemical and genomic data will be analyzed in relation to demographic data and outcome metrics. It is mentioned that “a large-scale genetic analysis has been conducted using the aforementioned NP phenotyping data”, and that “environmental risk models have been developed”. Please provide citations of related analyses/similar studies that support the use of these data.

DOLORISK Dundee is an important initiative that has the potential to provide longitudinal demographic and genetic information in neuropathic pain. Despite the above criticisms, I recognize the

	difficulty in analyzing large data sets, particularly when standardized forms are used for data collection, and I recognize that it is not feasible to collect specific information on pain phenotype and medication use. As such, the authors should aim to modify future methods of data collection, and subsequent publications should emphasize the limitations in the generalizability of their analyses.
--	--

REVIEWER	Dr. Maryam Shaygan Shiraz University of Medical Sciences, Iran
REVIEW RETURNED	09-Oct-2020

GENERAL COMMENTS	- The recently proposed definition of chronic pain by Treede et al., (2019) describes it as “pain that persists or recurs for longer than 3 months and is associated with significant emotional distress or functional disability (interference with activities of daily life and participation in social roles)”. Therefore, the criteria of “emotional distress or functional disability” in the definition of chronic pain should be considered. - According to IASP, “neuropathic pain is a clinical description (and not a diagnosis) which requires a demonstrable lesion or a disease that satisfies established neurological diagnostic criteria”. Thus, the presence of symptoms or signs (e.g., touch-evoked pain) alone does not justify the use of the term neuropathic. On the other hand, the authors have expressed that the study has reduced power to investigate aetiologies of pain in participants. This limitation may severely distort the results. - Give the eligibility (inclusion and exclusion) criteria of participants. - Please discuss the reliability and validity of the measures. - Please describe the answer choices of questions about locations and aetiologies of pain. - Please include a short explanation of the results regarding psychological variables such as depression, catastrophizing, etc.
--

VERSION 1 – AUTHOR RESPONSE

Reviewer: 1 (Michael Staudt)

Comments to the Author

The current article outlines the creation of a neuropathic pain database, DOLORISK Dundee, using two pre-existing Scottish population cohorts (GoDARTS and GS:SFHS). The GoDARTS cohort focused primarily on diabetic patients, whereas patients in GS:SFHS were recruited from general practice. From these cohorts, all diabetic patients from GoDARTS and all patients from GS:SFHS were mailed a questionnaire on neuropathic pain and other pain-related items. Neuropathic pain specifically was assigned based on responses to the DN4 and S-LANSS questionnaires. As this is a descriptive and not investigative article, the results outline the population demographics of those participants who responded and provided consent, as well as characteristics of those patients who completed 18-month follow-up.

Neuropathic pain is common and debilitating, however our understanding of its pathophysiology and natural history remain limited. One of the primary limitations is the lack of large-scale demographic data that collates validated outcome metrics and genomic data. DOLORISK Dundee is an exciting venture as it has the potential to longitudinally study a large neuropathic pain patient population with linkage to genetic association studies. This will hopefully allow collaboration with and comparison to similar population-based databases.

Thank you for these comments.

As this is a descriptive report of the baseline database information without formal analysis, my comments are limited to the nature of data collection and the potential future applications.

1. My primary criticism is the lack of data regarding pain characterization and etiology. Neuropathic pain is diverse, and includes multiple different mononeuropathies and polyneuropathies (both symmetric and asymmetric). Despite a consistent approach to overall pain phenotyping, there appears to be limited ability to specifically identify different neuropathic pain phenotypes. Diabetic neuropathy is not the same as CRPS which is not the same as lumbar radiculopathy, which can dilute interpretations following formal analysis. As neuropathic pain syndromes can be quite heterogeneous, this tends to be a problem in smaller studies – the DOLORISK Dundee database may have challenges of a different nature if interpretations are made based on overall “neuropathic pain”, which will limit generalizability to specific conditions. “Pain cause” is included in table 3, reported in approximately 2/3 of respondents, and it is mentioned that this is self-reported information from a list of common pain etiologies. It is important to include both this list of pain etiologies, as well as the specific breakdown in table 3.

The authors agree with the reviewer and recognise that neuropathic pain is a heterogeneous disease consisting of many aetiologies, each likely to have difference in pathophysiology. The challenge with large population cohorts is to capture this information in a way that is valid and informative for analysis, but also ensures adequate response rates from participants. Complex clinical phenotyping items are generally not feasible in self-complete questionnaires. Because of this, we used a relatively simple question asking about the cause of pain with common (neuropathic) pain aetiologies, to produce as high a response rate as possible. As noted, this approach is consistent with the international consensus on phenotyping neuropathic pain for population studies (NeuroPPIC; <https://doi.org/10.1097/j.pain.0000000000000335>). We have added the exact wording of the question and possible responses to table 4 (formerly table 3) in line with the reviewers suggestion. We have also added the following text to the manuscript (P11, L22-26):

“Whilst we attempted to obtain a broad array of common neuropathic pain aetiologies, it was not possible to capture detailed clinical information on neurological and sensory phenotypes, through the medium of a self-complete questionnaire.”

2. In addition to challenges in assessment based on individual phenotype, there are also limitations regarding the broader classification of central vs peripheral neuropathic pain. Although data are limited, previous studies have shown that patients with central pain overall do worse with medical management compared to peripheral pain. By combining these groups in one neuropathic pain cohort, this potentially dilutes generalizability of future findings.

Thank you for raising this point. Despite the limitations in conducting a self-report survey, outlined in the previous point, we are able to identify participants with neuropathic pain potentially caused by stroke and multiple sclerosis. This is through the “pain cause” question mentioned above (for which we have now added the exact wording and possible responses). Therefore we have the ability to compare some central neuropathic pain aetiologies to peripheral neuropathic pain aetiologies.

3. Conversely, having a large cohort of diabetic patients in GoDARTS can be seen as a strength when assessing diabetic neuropathy specifically, as up to a third of diabetic patients develop painful diabetic neuropathy. Again, although the nature of data collection does not mean that all patients in this cohort have painful diabetic neuropathy specifically, this would be an advantage in terms of analyzing demographic and genetic information in relation to treatment responses, and following long-term natural history.

Thank you for these comments. Although not all participants with neuropathic pain in GoDARTS will have diabetic neuropathy, we have included the Michigan Neuropathy Screening Instrument, to help identify those that do. We have added a sentence to the Strengths and Limitations section to convey this point (P11, L28-30):

“However, the use of GoDARTS as a diabetic cohort will strengthen analysis relating to diabetic neuropathy, particularly with the use of the MNSI as a screening tool.”

4. The question regarding medication intake is quite rudimentary (“Are you currently taking medications specifically to treat pain or discomfort?”), and the response rate of 95% or greater is not surprising. However, this response could be interpreted as patients taking the occasional ibuprofen all the way to regular and supervised use of methadone. In patients with neuropathic pain it is important to understand the classes of pain medications used (i.e. opioids, antidepressants, anticonvulsants, etc), and for opioids the morphine equivalent dose.

Thank you for raising this point. As the severity of pain can fluctuate in patients, the pain medication question was included in the baseline and follow-up surveys as an additional pain identification item in order to capture people whose pain had temporarily remitted (at the time of completing the questionnaire) and may answer ‘no’ to the question asking whether they are currently troubled by pain. This allows them to complete the rest of the questionnaire on the nature of their pain, including their average and worse pain over the last 3 months (as part of the chronic pain grade). The authors agree that more detailed questions are required when investigating treatment response and will consider this in future DOLORisk surveys. A sentence has been added to the manuscript to clarify the above point (P8, L24-26).

“The second question was added to allow inclusion of anyone whose pain was temporarily relieved by analgesics at the time of completing the questionnaire and may therefore answer ‘no’ to the preceding question.”

5. I did not find any information regarding the use of surgery (i.e. neuromodulation) in the treatment of neuropathic pain. This is also important information to discern in the assessment of long-term treatment outcomes. Future surveys should incorporate relevant questions.

Thank you for bringing this to our attention. Whilst participants could indicate that they were currently taking medications to treat pain, we did not ask about other forms of treatment such as neuromodulation therapy. We have added this as a limitation of the current cohort and will consider these questions in future surveys (P12, L25-28).

“Second, whilst the participants were asked about whether they were currently taking medications to treat pain, they were not asked about other forms of pain treatment, such as neuromodulation therapy. These are important aspects when assessing pain outcomes and will be considered in future surveys.”

6. The authors should indicate how biochemical and genomic data will be analyzed in relation to demographic data and outcome metrics. It is mentioned that “a large-scale genetic analysis has been conducted using the aforementioned NP phenotyping data”, and that “environmental risk models have been developed”. Please provide citations of related analyses/similar studies that support the use of these data.

We are conducting two separate analyses. The first is a genome-wide association study (GWAS) meta-analysis, from which polygenic risk scores will be developed to provide a genetic model for neuropathic pain. We have previously conducted two GWAS studies in GoDARTS using a prescription-based phenotype and polygenic risk scores have been used to great effect in type 2 diabetes (<https://doi.org/10.1002/ejp.560>; <https://doi.org/10.1016/j.ebiom.2015.08.001>; <https://doi.org/10.1038/s41588-018-0241-6>). The second is the development of environmental prediction models using a combination of demographic, clinical, psychological, lifestyle and biochemical data. The methodology for building the models will be based on previous studies published in low back and postsurgical pain (<https://doi.org/10.1371/journal.pmed.1002019>; <https://doi.org/10.1097/j.pain.0000000000001945>). We have conducted a recent review of both genetic and environment factors of neuropathic pain and it is based on this that we have included them in our analysis (<https://doi.org/10.1097/j.pain.0000000000001824>).

We then plan to combine the genetic and environmental models to ascertain their relative contributions to the onset of neuropathic pain.

We have updated the manuscript to reflect this (P10, L22-31; P11, L1-6):

“In addition to the descriptive characteristics of DOLORisk Dundee, a genome-wide association study meta-analysis has been conducted using the aforementioned NP phenotyping data (combined with the UK Biobank cohort using a prescription-based phenotype). Similar genetic studies have been conducted for neuropathic pain using a prescription-based phenotype in GoDARTS^{29 30}. Separately, environmental risk models have been developed, incorporating demographic, clinical, and biochemical predictors available through pre-existing records (table 5) and psychological and lifestyle data collected through the baseline survey. Both genetic and environmental predictors have been identified in previous studies of neuropathic pain³¹. This will use similar methodologies employed in low back and postsurgical pain^{32 33}. The results of these analyses will be submitted for publication in the near future. Eventually the environmental and genetic analyses will be combined through the use of polygenic and environment risk scores to produce a joint genetic and environmental model that will enable better understanding of the relative contributions of genetic and environmental factors to the variation in NP onset and progression. Polygenic risk scores have already been explored to good effect in type 2 diabetes³⁴.”

DOLORISK Dundee is an important initiative that has the potential to provide longitudinal demographic and genetic information in neuropathic pain. Despite the above criticisms, I recognize the difficulty in analyzing large data sets, particularly when standardized forms are used for data collection, and I recognize that it is not feasible to collect specific information on pain phenotype and medication use. As such, the authors should aim to modify future methods of data collection, and subsequent publications should emphasize the limitations in the generalizability of their analyses. The authors would like to thank the reviewer for these comments and will take them into consideration for future publications.

Reviewer: 2 (Dr. Maryam Shaygan)

Comments to the Author

- The recently proposed definition of chronic pain by Treede et al., (2019) describes it as “pain that persists or recurs for longer than 3 months and is associated with significant emotional distress or functional disability (interference with activities of daily life and participation in social roles)”.

Therefore, the criteria of “emotional distress or functional disability” in the definition of chronic pain should be considered.

Thank you for this suggestion. We think that the definition given by the reviewer is for “chronic primary pain”, rather than for “chronic pain”. In the paper to which the reviewer probably refers (which was co-authored by the senior author of this paper), the statement is, “Chronic pain is defined as pain that persists or recurs for more than 3 months.” The aim of DOLORisk Dundee is to build risk models for neuropathic pain. Because of this we included a number of items in the questionnaire assessing emotional distress and functional disability including the EQ5D (health related quality of life), depression, anxiety, sleep disturbance and the chronic pain grade (which includes aspects on pain interference). Since we are trying to understand what factors (genetic and environmental) predict

neuropathic pain onset and psychological and functional disability have been identified in previous studies, these factors have been included in our modelling, which precludes their use as a definition criteria.

- According to IASP, “neuropathic pain is a clinical description (and not a diagnosis) which requires a demonstrable lesion or a disease that satisfies established neurological diagnostic criteria”. Thus, the presence of symptoms or signs (e.g., touch-evoked pain) alone does not justify the use of the term neuropathic. On the other hand, the authors have expressed that the study has reduced power to investigate aetiologies of pain in participants. This limitation may severely distort the results. This point is similar to that raised by reviewer 1. We agree with the reviewers that this is a limitation and we have attempted to acknowledge this in the manuscript (please see point 1 above). In addition, we have used the Neuropathic Pain Phenotyping by International Consensus (NeuroPPIC) criteria, which specifies an entry-level basis for phenotyping “possible neuropathic pain”. These criteria include 1. Symptom assessment using a validated neuropathic pain screening tool, 2. Anatomical distribution of pain using a checklist, 3. History of pain including intensity, duration and relevant pain pathologies and demographic information. We have therefore added the term “possible” to the manuscript (P10, L15; L18; 20), based on the IASP Grading System for neuropathic pain (<https://doi.org/10.1097/j.pain.0000000000000492>).

- Give the eligibility (inclusion and exclusion) criteria of participants.

We have added a table providing the inclusion and exclusion criteria of the study to the manuscript and have referenced this in the text.

- Please discuss the reliability and validity of the measures.

We have added the following text to the manuscript (P9, L3-7).

“Both questionnaires are screening tools for NP and have been validated for use in populations with neuropathic pain. Both screening tools have also been validated as self-report items and are therefore suitable for use in a postal/online survey setting. The DN4 showed 78% sensitivity and 81% specificity, whilst the S-LANSS had 74% sensitivity and 76% specificity when compared to clinical examination.”

- Please describe the answer choices of questions about locations and aetiologies of pain.

This is a similar point to that raised by reviewer 1 in point 1. We have added the question and answer choices for both the pain location and cause to table 4.

- Please include a short explanation of the results regarding psychological variables such as depression, catastrophizing, etc.

We have added the following text to the manuscript (P8, L15-22):

“A summary of the results from respondents of these psychosocial items are given in table 5, dichotomised by cohort. When compared to GS:SFHS, GoDARTS had a higher mean score in all of the psychological aspects (where a higher score denotes more of the concept being measured), including depression, anxiety, sleep disturbance and pain catastrophizing, as well as having a lower mean score in the five personality traits (extraversion, agreeableness, conscientiousness, emotional stability and open to new experiences) and health-related quality of life.”

P9, L14-17:

“When compared to GS:SFHS, GoDARTS had a higher mean score in both the DN4 and S-LANSS, potentially reflecting the older age of GoDARTS and the diabetic nature of the cohort (table 5).”

VERSION 2 – REVIEW

REVIEWER	Staudt, Michael Michigan Head and Spine Institute, Neurosurgery
REVIEW RETURNED	Michigan Head and Spine Institute, Neurosurgery 27-Dec-2020

GENERAL COMMENTS	The authors present a revised version of their manuscript regarding the creation of a neuropathic database, DOLORISK Dundee. The authors have satisfactorily addressed this reviewer's comments and expanded their manuscript accordingly. Both reviewers commented on the heterogeneity of neuropathic pain and the difficulty in capturing these data in large patient cohorts, which has been appropriately clarified and expanded upon. I also appreciate the updated discussion on the utilization of biochemical and genomic data. At this time, I recommend acceptance. I eagerly future publications utilizing these data that evaluate longitudinal neuropathic pain cohorts linked with large-scale genetic data.
---